# Preliminary Evaluation of Pentacene Field Effect Transistors with Polymer Gate Electret as Ionizing Radiation Dosimeters

Irina Valitova [1], Alexandria Mitchell [1], Michael A. Hupman [1], Ian G. Hill [1] and Alasdair Syme [1,2,3,*]

1 Department of Physics and Atmospheric Science, Dalhousie University, Halifax, NS B3H4R2, Canada; Irina.Valitova@dal.ca (I.V.); Lexie.Mitchell@dal.ca (A.M.); Allan.Hupman@Dal.Ca (M.A.H.); Ian.Hill@Dal.Ca (I.G.H.)
2 Department of Radiation Oncology, Dalhousie University, Halifax, NS B3H 4R2, Canada
3 Department of Medical Physics, Nova Scotia Health Authority, QEII Health Science Centre, Halifax, NS B3H 1V7, Canada
* Correspondence: alasdair.syme@nshealth.ca

**Abstract:** Interest in the use of organic electronic devices in radiation sensing applications has grown in recent years. The numerous device configurations (e.g., diodes, thin film transistors) and potential for improved tissue equivalence compared to their silicon-based analogues make them attractive candidates for various radiation dosimetry measurements. In this work, a variation of the organic thin film transistor (OTFT) is studied, in which a polymer electret is added. An OTFT electret design can be used in either a wired or wireless configuration for in vivo dosimetry with the possibility of real-time detection. The linearity, reproducibility, and dependence on energy of these devices were measured through exposure to 100 kVp photons from an orthovoltage treatment unit (Xstrahl 300) and 6 MV photons from a Varian TrueBeam medical linear accelerator. Prior to irradiation, all transistors were programmed with a $-80$ V bias applied to the Gate electrode ($V_g$) for 3 s. In the wireless configuration, after each delivered dose, the transfer characteristic was scanned to readout the amount of erased charges by monitoring the drain current change. When the programmed charge was sufficiently depleted by radiation, transistors were reprogrammed for repeated use. The real-time readout in a wired configuration was performed by measuring the drain current with $V_g = -15$ V; $V_d = -15$ V. The 6 MV photon beam was turned on and off at different dose rates of 600, 400, 300, 200, and 60 cGy/min to quantify the sensitivity of the device to changes in dose rate. The wireless transistors showed a linear increase in current with increasing dose. The sensitivities for different energies were $60 \pm 5$ nA/Gy at 6 MV at a dose rate of 600 cGy/min and $80 \pm 10$ nA/Gy at 100 kVp at a dose rate of 200 cGy/min. The sensitivity of detectors tested in a wired configuration at $V_d = -15$ V; $V_g = -15$ V was 8.1 nA/s at a dose rate of 600 cGy/min. The principle of pentacene OTFTs with polymer electret as radiation detectors was demonstrated. Devices had excellent linearity, reproducibility, and were able to be reprogrammed for multiple uses as wireless detectors. The wired transistors demonstrated an effective response as real-time detectors.

**Keywords:** organic semiconductor detector; organic thin film transistors with polymer electret; radiation; radiation dosimetry

## 1. Introduction

Organic semiconductor dosimeters have been introduced recently [1–12]. Organic thin film transistors (OTFT) have gained special interest in radiation dosimetry applications as a new class of potentially tissue-equivalent radiation dosimeters. The sensitivity of such dosimeters is generally defined as the ratio of the channel conductance change to the absorbed dose and parameters, such as off current, on current, threshold voltage $V_{th}$, and current ratio, can be monitored [13–16]. In a bottom-gate top-contact design with poly(3-hexylthiophene-2,5-diyl) (P3HT) as a semiconductor on a silicon dioxide ($SiO_2$) dielectric layer, Raval et al. found a decrease in the on current by a factor of two, an increase

in off current by a factor of 150, and a decrease in the mobility after a total dose of 41,000 cGy. The threshold voltage was shifted negatively due to positive charge accumulation in the silicon dioxide [13]. In a pentacene OTFT, after exposure to a total dose of 100 Gy, the off current was increased 320 times, which resulted in a sensitivity of 20 nA/Gy. The threshold voltage shift resulted in a sensitivity of 0.3 V/Gy [14]. Furthermore, the same group introduced CuPc OTFT dosimeter after $\gamma$-irradiation, with a minimum dose of 10 cGy, increasing to a maximum dose of 100,000 cGy. Three and five OTFTs, respectively, were connected in parallel to increase the resolution of measuring the off current. The measured sensitivity from off current shifts after irradiation was of 0.02/cGy for $\Delta I_{OFF}/I_{OFF}$. From the $V_{th}$ shift measured at a constant drain current of $10^{-7}$ A, the sensitivity of $1.5 \times 10^{-4}$/cGy for $\Delta V_{TH}/V_{TH}$ was observed for a total dose of 100,000 cGy [15].

Kim et al. introduced a rubrene semiconductor OTFT as a dosimeter for electron beam irradiation [17]. They showed that electrons can induce traps not only in $SiO_2$ dielectric and $Si/SiO_2$ interface, but also in the organic semiconductor. They found a significant mobility change after irradiation of all layers of transistors and negligible change of mobility after irradiation of dielectric only (before rubrene deposition), which proves that radiation mainly affects the organic semiconductor. Basirico and colleagues only partially related the reduction of charge carrier mobility with an increase of charge carrier traps in 6,13-bis(triisopropylsilylethynyl)-pentacene (TIPS-pentacene)-based field effect transistors with organic dielectric irradiated by high energy protons [18]. The high-energy protons induce defects in the organic dielectric and strains in the TIPS-pentacene layer leading to mobility reduction.

The sensitivity of OTFT dosimeters can be improved by introducing additional carrier traps to the organic polymer or dielectric material. This distribution of traps within the band gap are associated with donor and acceptor atoms, with other impurity atoms in the bulk of the semiconductor, or with "dangling bonds" that occur at defects or exterior surfaces and grain boundaries. Jain et al. improved the response of OTFTs to radiation by increasing the density of trap states by introducing impurity atoms into the bulk of the semiconductor. The TIPS-Pentacene and Polystyrene (TP-PS) blend was used as a semiconductor material in OTFTs for sensing gamma rays from a cobalt-60 ($^{60}$Co) radiation source [19]. Devices irradiated before deposition of the TP-PS (only dielectric layer) and devices irradiated after deposition of the semiconductor were compared to show the amount of charge trapped in the bulk semiconductor (TP-PS blend) and in the $SiO_2$/TP-PS interface. The sensitivity of the $SiO_2$/TP-PS detectors was significantly higher than the sensitivity of detectors irradiated before TP-PS deposition [19].

The working mechanism of OTFT dosimeters is based on the trapping of photogenerated charges on the interface or within the organic semiconductor upon X-ray exposure. To increase the sensitivity of OTFT dosimeters, we explore the use of organic memory devices, such as polymer electret OTFTs, where additional charges are stored within the bulk of the dielectric film and at the interface between gate dielectrics and dielectric/semiconductor channels [20–23]. In a polymer electret, the channel conductance is modulated by the electrostatically trapped charges inside the polymer electret upon applying a programming or erasing gate voltage. Programming of the polymer electret provides a built-in electric field in the dielectric of the device. Radiation-generated carriers within the dielectric recombine with trapped charges, reducing the built-in field.

Although dosimeters based on the electret principle are not new, published literature in this area has focused on gas-filled, ion chamber-like devices [24–27]. To the best of our knowledge, there are no studies dealing with dosimeters based on an OTFT electret design. An advantage of dosimeters based on an OTFT electret design is the ability to program it for use in either a wired or wireless configuration for in vivo dosimetry. Wired devices permit real time detection, whereas wireless devices are significantly more convenient for clinical use. Additionally, such devices can be reprogrammed for repeated use. In this work, we describe the design, fabrication, and testing of pentacene OTFTs with a polystyrene electret gate on a silicon substrate for preliminary evaluation as ionizing radiation dosimeters. We

propose a mechanism for charging a polymer electret and demonstrate wireless and wired detection of 100 kVp and 6 MV energies X-rays, produced with an orthovoltage X-ray unit and a medical linear accelerator, respectively.

## 2. Materials and Methods

### 2.1. Fabrication of Pentacene Transistors

N-type, heavily As-doped Si (100) wafers (0.001 Ohm-cm) with 300 nm thermal $SiO_2$ were washed with deionized water, acetone, and ethanol; dried with an air gun; and treated in a UV-ozone cleaner for 20 min. The polystyrene (Sigma-Aldrich, Saint Louis, MO, USA, Mw = 280 kg/mol) was used as purchased to make a solution of 0.5% $w/v$ in toluene (99.5%, Sigma-Aldrich, Saint Louis, MO, USA). The PS solution was spin coated at 6000 rpm for 1 min and subsequently annealed on the hot plate at 90 °C for 60 min, forming a second, 45 nm dielectric layer on top of the $SiO_2$.

The pentacene (TCI America, Portland, OR, USA) was thermally evaporated through a shadow mask onto the PS layer at $10^{-6}$ mbar vacuum at a rate of 1 A/s, with the substrate held at 50 °C during evaporation to a thickness of 50 nm. The source and drain interdigitated Au electrodes were thermally evaporated through a shadow mask onto the pentacene layer to a thickness of 50 nm at a rate of 1 A/s.

Finally, devices were encapsulated with an evaporated Parylene C layer (1 μm). For a 1 um of Parylene-C evaporation, 1 g of Parylene-C was placed into the deposition system and left until all of the material was deposited.

### 2.2. Electrical Characterization

Capacitance of a Si/$SiO_2$ PS/Au capacitor was measured using a 4274 A multi-frequency LCR meter. For electrical characterization of the OTFTs, a dual channel SMU (Keithley 2614B, Tektronix, Inc., Beaverton, OR, USA) was used. All transistors were placed in a sample holder that was 3D-printed with polylactide (PLA). Metal spring pins were used for electrical contacts, and a triaxial cable was used to connect the devices to the SMU, which was housed outside of the treatment vaults.

### 2.3. Irradiation Experiments

All devices were irradiated with 6 MV photon beams on a Varian TrueBeam (Varian Medical Systems, Inc., Palo Alto, CA, USA) medical linear accelerator and with a kilovoltage (100 kVp) photon beam on an Xstrahl 300 orthovoltage X-ray unit (Xstrahl Ltd., Surrey, UK).

For 6 MV irradiations, the transistors in the 3D-printed sample holder were placed at isocenter with a $5 \times 5$ cm$^2$ field size on top of a 15 cm stack of Solid Water (Sun Nuclear Corp., Melbourne, FL, USA) for backscatter and below an additional 5 cm of Solid Water, as shown in Figure 1a. A PLA frame with a cap was used to protect the device from physical compression by the Solid Water and to minimize the air gap ($\approx$1 mm) between the device and the solid water placed on top. The source-to-detector distance was set to 100 cm for all irradiations. The output of the accelerator under these measurement conditions was verified using a calibrated ion chamber. Devices were exposed to doses ranging from 1 to 10 Gy of 6 MV photon beams.

In the orthovoltage unit an applicator cone of 5 cm diameter, 30 cm length was used to deliver dose directly to the diode. Diodes were placed on top of a 15 cm stack of Solid Water for backscatter as shown in Figure 1b. These measurement conditions differed from the 6 MV beam due to the less penetrating nature of the 100 kVp photons compared to the 6 MV photons. Doses between 1 and 30 Gy were delivered.

For wireless readouts, all transistors were programmed with a $-80$ V bias applied to the Gate electrode ($V_g$) for three seconds at $V_d = 0$ V to write the charges to the polymer electret. Transistors were then irradiated, and after each delivered dose, the transfer characteristic was scanned to readout the drain current change at a fixed bias to quantify changes in the I–V curve under these specific conditions. When programmed charges were depleted by radiation, transistors were reprogrammed for repeated use.

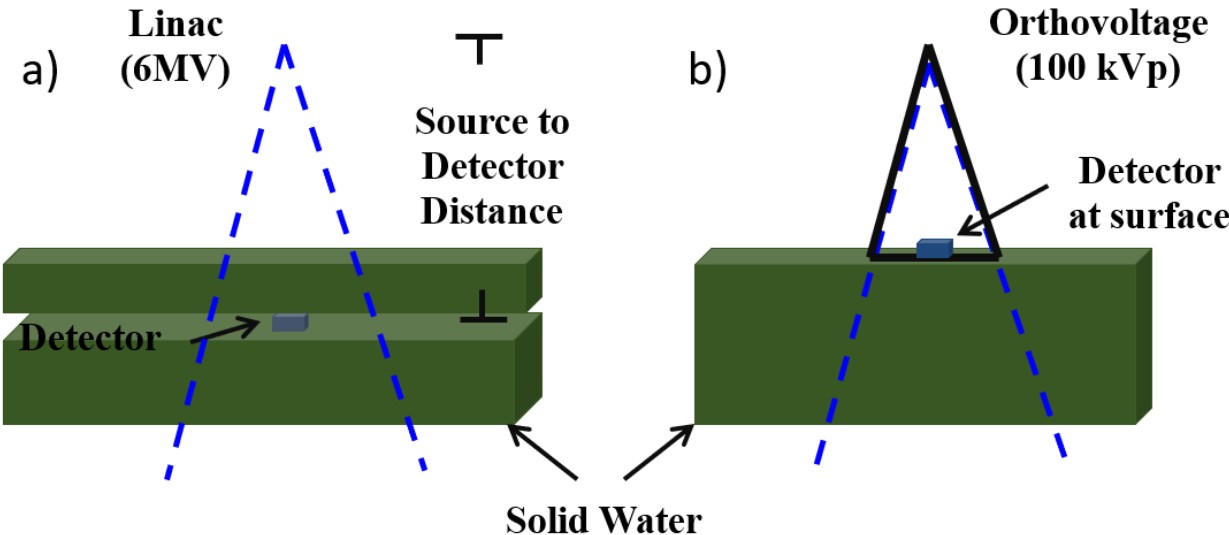

**Figure 1.** A schematic representation of the setup used to irradiate the OTFT detector with (**a**) a 6 MV photon beam and (**b**) a 100 kVp photon beam. Measurements were performed in Solid Water to mimic a clinical setup and allow us to relate the response of the detector to a known dose in water. An ion chamber was used to calibrate both treatment machines. The dashed blue lines represent the projected shape of the radiation field. The black triangle in (**b**) represents the cone that is physically attached to the orthovoltage X-ray unit and helps collimate the field.

The real-time readout in a wired configuration was conducted after programing with a $V_g = -80$ V bias for three seconds. The drain current was measured at an electric field at $V_g = -15$ V; $V_d = -15$ V. After the drain current was stabilized, the 6 MV photon beam was turned on and off at different dose rates of 600, 400, 300, 200, and 60 cGy/min for each step.

### 3. Results

*3.1. Electrical Characterisation*

Pentacene OTFTs were fabricated with a bottom-gate top-contact configuration as shown in Figure 2. A highly doped n type Si wafer was used as the substrate and gate electrode, a 300 nm $SiO_2$ as a gate dielectric, and approximately 45 nm of polystyrene as a polymer electret. The thickness of Polystyrene was estimated from the capacitance value and was in good correlation with thickness measured on profilometer (48 nm). The capacitance ($C_{tot}$) of the $SiO_2$/PS dielectric was 3.26 nF at 1 kHz; the capacitance of the Polystyrene layer was 45.45 nF, as determined from the capacitance expression (1):

$$\frac{1}{C_{tot}} = \frac{1}{C_{SiO_2}} + \frac{1}{C_{PS}} \tag{1}$$

Figure 3 shows the output and transfer characteristics of the Pentacene transistors. Output characteristics were scanned at $V_d$ = from 0 to $-15$ V at $V_{gs}$ = 0 V, $-5$ V, $-10$ V, and $-15$ V. Transfer characteristics were scanned at $V_g$ = from 0 to $-15$ V, at a constant $V_d = -15$ V. On/Off ratios of transistors were typically $10^3$, and a threshold voltage of $V_{th} = -8.4$ V was extrapolated from $\sqrt{I_d}$ vs. $V_g$ curves at $V_d = -15$ V. The charge carrier mobility $\mu_{sat} = 0.6$ cm$^2$V$^{-1}$s$^{-1}$ was derived from $I_d$ in the saturation regime at $V_d = -15$ V using Equation (2):

$$\mu_{sat} = \frac{2L}{WC}\left(\frac{d\sqrt{I_{dsat}}}{dV_g}\right)^2 \tag{2}$$

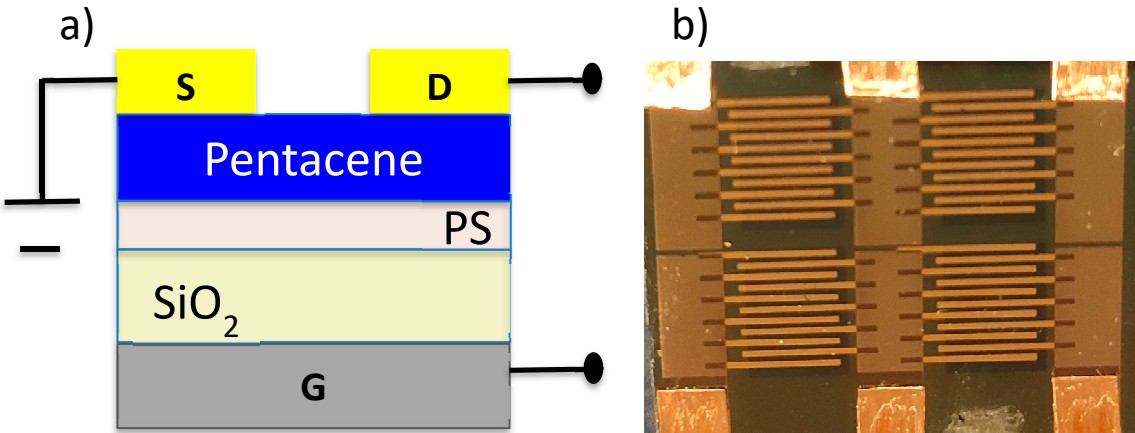

**Figure 2.** (**a**) Schematic configuration of the pentacene-based OTFT electret detector. (**b**) The top view of four transistors with interdigitated electrodes.

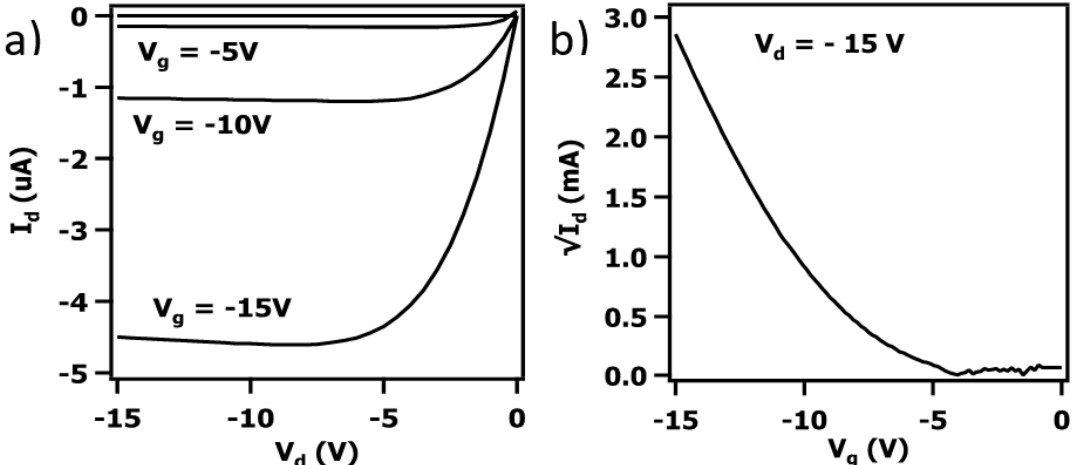

**Figure 3.** (**a**) Output characteristics of pentacene OTFTs with PS as the polymer electret at $V_g$ = 0 V, −5 V, −10 V, and −15 V; (**b**) Transfer curve of the pentacene-based OTFTs with PS electret at $V_d$ = −15 V.

The charge carrier mobility $\mu_{lin}$ = 0.79 cm$^2$V$^{-1}$s$^{-1}$ was derived from $I_d$ in the linear regime at $V_d$ = −1 V using Equation (3):

$$\mu_{lin} = \frac{L}{WCV_d} \frac{dI_d}{dV_g} \tag{3}$$

### 3.2. Programming and Stability

To charge the polymer electret, a −80 V bias was applied to the Gate electrode for three seconds at $V_s$ = $V_d$ = 0 V, as shown in Figure 4a. The observed change in transistor characteristics is consistent with hole trapping in the PS layer. The trapped holes on the interface with pentacene induce the more negative shift of the threshold voltage. The small difference in the HOMO energy level between pentacene and PS facilitated the transfer of holes from pentacene to the polymer electret. [21] Figure 4b shows the stability of the drain current after programming with −80 V gate bias for 3 s. The drain current was measured at $V_d$ = −15 V and $V_g$ = −15 V. The data indicate that incorporated charges persist for a length of time that is sufficient to permit the use of devices in a clinical setting.

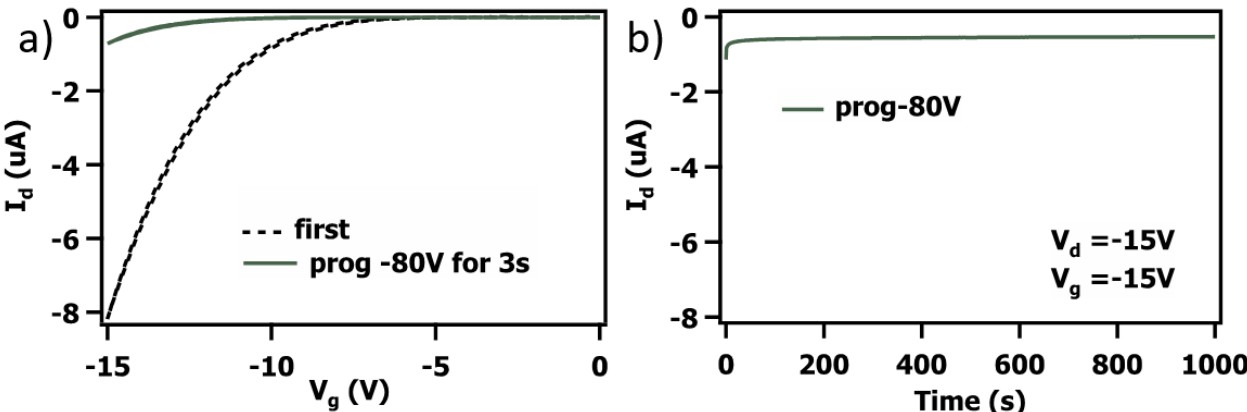

**Figure 4.** (**a**) Transfer curves of the pentacene-based OTFTs with PS electret before and after programming by applying the −80 V gate bias for 3 s; (**b**) the stability of the drain current after programming with negative −80 V gate bias for 3 s measured at $V_d$ = −15 V, $V_g$ = −15 V. 'First' refers to the I–V curve in the absence of programming and 'prog' refers to the I–V curve following the programming at −80 V for 3 s.

The retention time of the pentacene OFETs after programing with gate pulse $V_g$ = −80 V is shown in Figure 4b. The programming is stable on the timescales of the experiments and is maintained for at least 1000 s. The drain current was measured at $V_d$ = −15 V and $V_g$ = −15 V. This level of stability would be considered acceptable for most clinical in vivo dosimetry applications.

### 3.3. Irradiation Response and Linearity

Generally, X-ray photon interactions lead to the creation of Compton electrons, photo-electrons, or electron-positron pair production, which in turn create electron-hole pairs and phonons. Under the influence of an applied electric field, the electron-hole pairs separate and drift to the respective electrodes, and on the way, can recombine with the trapped charges, leading to a shift in threshold voltage toward the positive direction, as shown in Figure 5a. This behaviour is accompanied by an increase in drain current at a fixed gate voltage of −15 V because of the shift induced in the IV curve. Therefore, the shift in drain current and/or threshold voltage can be considered as radiation dose metrics. In this work, the drain current shift at $V_g$ = −15 V is used as a sensing metric, as shown in Figure 5. In a wireless readout configuration, devices were programmed and scanned then irradiated and scanned again to check the current shift in response to doses between 1 and 15 Gy (in 1 Gy increments). The data shown in Figure 5b indicate that the response is linear, following irradiation with a 100 kVp beam. The linearity of transistor response was also observed at variable dose steps (1 Gy to 6 Gy per step) up to an accumulated dose of 29 Gy, as shown in Figure 5c,d. These devices, like many solid state radiation sensors, exhibit a change in sensitivity at low dose levels which then stabilize; thus, the 1 and 2 Gy points have not been included in the fit for this reason.

The device sensitivity can be expressed as the change in drain current per unit dose: $S = \Delta I_d / D$. Figure 6 shows the $I_d$ vs. dose relationship following a duplicate series of programming and irradiation steps. The data demonstrate the potential of these devices to be reused across multiple irradiations. The sensitivity of the device used in the 6 MV beam remained essentially unchanged. The device used for the 100 kVp irradiations showed a small change in sensitivity, though it preserved its linear response with dose.

After irradiation with a kilovoltage 100 kVp photon beam and megavoltage 6 MV photon beam, programed charges were depleted. Some devices were then reprogrammed for additional experiments. For the wireless readout configuration in the megavoltage photon beam, the average sensitivity of the devices studied was approximately 60 ± 5 nA/Gy. At 100 kVp, the sensitivity of the devices was approximately 80 ± 10 nA/Gy.

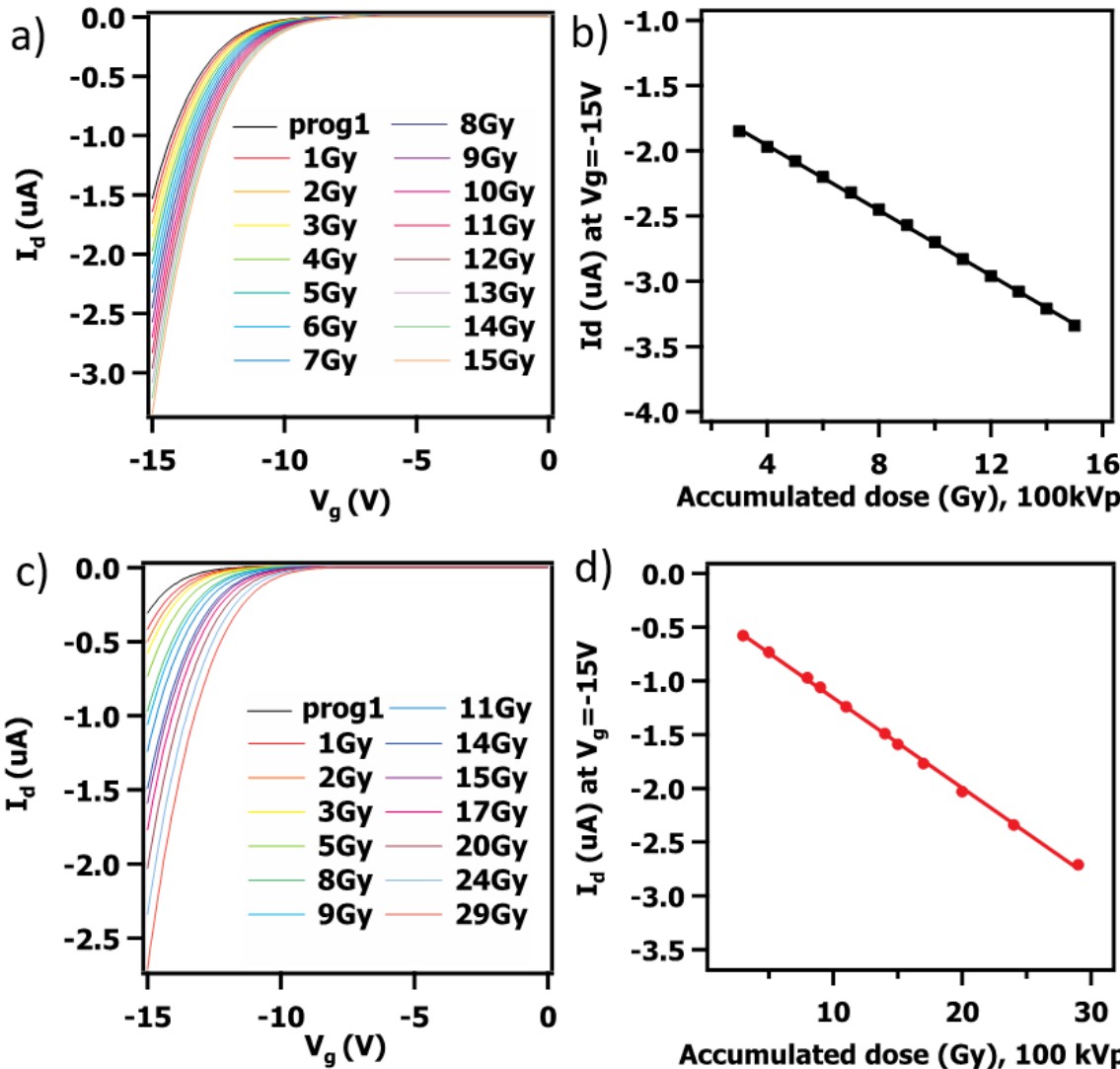

**Figure 5.** Transfer characteristics of pentacene OTFTs with PS electret after first programming and consequent accumulated irradiation with (**a**) 1 Gy steps and (**c**) variable dose steps. (**b**,**d**) The corresponding drain current value at $V_g = -15$ V as a function of accumulated dose. Irradiations were performed with a 100 kVp photon beam. The 1 and 2 Gy points have been excluded in the fit, as the sensitivity was more variable at these low doses. 'Prog1' refers to the I–V curve following the programming at −80 V for 3 s.

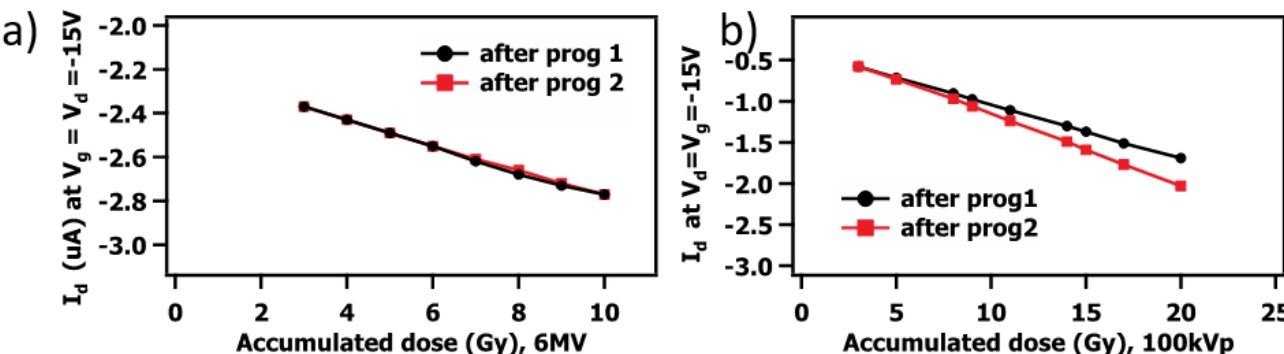

**Figure 6.** $I_d$ vs. dose for a device across two cycles of programming and irradiation at both (**a**) 6 MV and (**b**) 100 kVp. The sensitivity remained constant beyond doses of 3 Gy.

The pentacene OTFTs with polymer electret were also investigated for wired real-time readout of the 6 MV photon beam. After programing with a $V_g = -80$ V bias for three seconds, the drain current was measured at $V_d = -15$ V and $V_g = -15$ V. Figure 7a demonstrates the response of the detector at the electric field at $V_d = -15$ V and $V_g = -15$ V, when the radiation beam was turned on and off with variable dose rates during the beam-on times of 600, 200, 300, 400, and 60 cGy/min for each step, respectively. Figure 7b represents the corresponding slope of the current measured when the radiation beam was on.

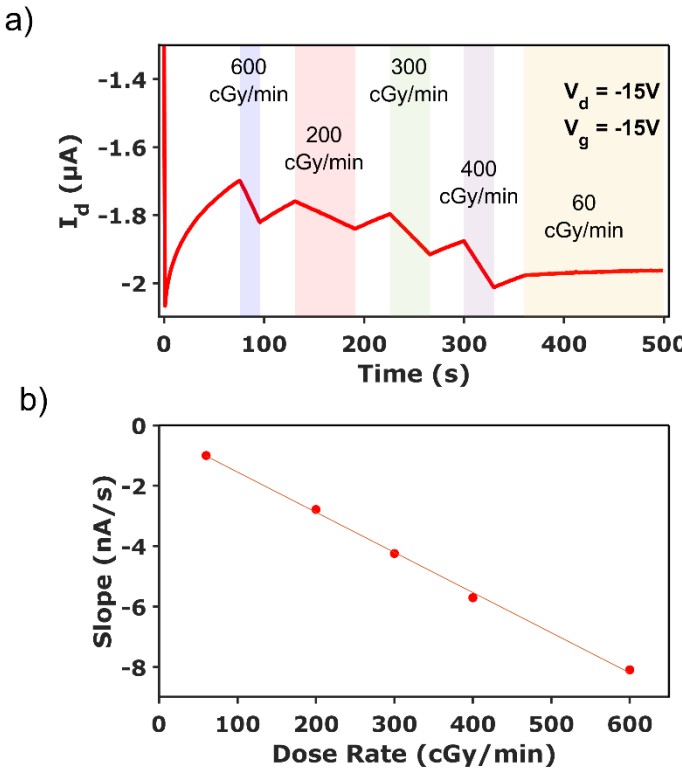

**Figure 7.** Real-time readout of a 6 MV photon beam after programing with a negative $V_g = -80$ V bias for three seconds; (**a**) the drain current was measured at $V_d = -15$ V and $V_g = -15$ V with radiation on and off with dose rates of 600 cGy/min, 200 cGy/min, 300 cGy/min, 400 cGy/min, and 60 cGy/min. (**b**) The corrected slope values of the current when the radiation beam was on at corresponding dose rates.

The sensitivity of detectors during real-time readout was determined as the slope of the current-time plot when the radiation was on. The data are shown in Table 1. At this electric field, the drain current did not stabilize and continued to shift negatively as a result of charging of the electret (i.e., programming continued during the experiment). The slope values (in units of nA/s) reported in Table 1 were corrected with the average value of the slope 10 s before the beam was on and after the beam was turned off to determine the net change in slope caused by irradiation. All devices showed an approximately linear dependence on the dose rate range from 60 cGy/min to 600 cGy/min, with an average sensitivity of devices from 1.0 nA/s to 8.1 nA/s, respectively, at $V_d = -15$ V and $V_g = -15$ V.

**Table 1.** The sensitivity of the pentacene OTFTs with PS electret during real-time readout in a wired configuration irradiated with a 6 MV photon beam.

| Dose Rates | Sensitivity (nA/s) at $V_{ds} = -15$ V; $V_{gs} = -15$ V |
|:---:|:---:|
| 600 cGy/min | 8.1 |
| 400 cGy/min | 5.7 |
| 300 cGy/min | 4.2 |
| 200 cGy/min | 2.8 |
| 60 cGy/min | 1.0 |

## 4. Discussion

Electrets are insulating materials with a quasi-permanent electric polarization once the incorporated charge persists for $\approx 10^9$ s [28]. To charge the polymer electret used as a second dielectric layer in OTFT, a $-80$ V bias was applied to the Gate electrode. The positive charges are induced from pentacene to the polymer electret and trapped in the electret. The trapped holes near the interface with pentacene induce a more negative shift of the threshold voltage. Radiation can erase the trapped charges and shift the threshold voltage back to the original state.

The sensitivity of OTFTs with PS electret dosimeters was checked in wireless and wired readout configurations. For the wireless readout configuration, the average sensitivity of the devices was $60 \pm 5$ nA/Gy for the 6 MV beam and $80 \pm 10$ nA/Gy for the 100 kVp beam. This was potentially due to the influence of high Z electrodes and a silicon wafer introducing an enhancement to the photon interactions that would not be present in a water medium (the medium in which the dose is being reported). At lower photon energies where the cross section for photoelectric interactions increases rapidly, the influence of high Z metal electrodes and non-tissue equivalent substrate on the collected current may not be negligible [10–12]. In a wired configuration, Compton current can be induced in the wires to increase the signal in a way that might not be proportional to the dose in the active volume of the detector. That process will not happen in the wireless configuration since the current is not collected while the beam is on. However, the presence of the Si wafer and the gold electrodes may induce a signal in the active volume of the device that will not be proportional to the dose in tissue because of the relative enhancement of the photoelectric cross section. Future work will explore the fabrication of these devices on organic substrates and use of conducting organic polymers as electrodes to determine if energy dependence can be reduced.

For the wired, real-time readout, the device demonstrated a linear response with dose rate over a range of 60 cGy/min to 600 cGy/min, with the sensitivity ranging from 1.0 nA/s to 8.1 nA/s, respectively at $V_d = -15$ V and $V_g = -15$ V.

Electron-hole pairs created after X-ray photon interactions must be separated and drift to certain electrodes to be able to erase the trapped charges. Therefore, the dosimeter response to radiation can be higher due to the better charge separation in a higher electric field and smaller in a lower electric field. Applying higher biases during readout, however, may perturb the programming of the device. Future work will investigate the optimization of the programming and readout cycles of these devices to maximize sensitivity and reproducibility.

## 5. Conclusions

OTFTs with an electret have demonstrated potential in radiation dosimetry applications. These devices in a wireless configuration possess excellent linearity over doses between 1 to 30 Gy at both kilovoltage and megavoltage X-ray energies, with sensitivities of 60–80 nA/Gy as the change in drain current at $V_g = -15$ V. For wired real-time readout, the sensitivity of the devices was around 8.1 nA/s at a dose rate of 600 cGy/min and electric

field at $V_d = -15$ V and $V_g = -15$ V. The devices are well-suited to in vivo dosimetry applications in radiation therapy. Advantages of the design include: options for both wireless and wired readout configurations, repeated readout due to signal storage, and repeated use due to the ability to reprogram the device. Moreover, these organic devices can be fabricated on flexible substrates with an organic dielectric to contain a tissue-equivalent composition. A similar atomic number between the dosimeter and human tissue will enhance precision and facilitate high quality measurement of patient doses. Future work will include the quantification of signal fade post-exposure, lifespan of the devices, and sensitivity improvement. For example, the sensitivity of the dosimeters can potentially be improved by increasing the trap carrier density in organic semiconductor or dielectric layer. Similar to organic memory devices, the dielectric layer can be complex or with a floating gate to store more charges and thus show the improved response to ionizing radiation.

**Author Contributions:** Conceptualization I.V., I.G.H. and A.S.; methodology: I.V., A.M. and M.A.H.; formal analysis: I.V. and A.M.; investigation: I.V. and A.M.; writing—original draft: I.V. and A.M.; writing—review and editing: M.A.H., I.G.H. and A.S.; project administration: I.G.H. and A.S.; funding acquisition: A.S. All authors have read and agreed to the published version of the manuscript.

**Funding:** This work was supported in part by a grant from the Natural Sciences and Engineering Research Council of Canada (Grant # RGPIN-2021-03715).

**Data Availability Statement:** Data is available from the authors upon reasonable request.

**Conflicts of Interest:** The authors declare no conflict of interest.

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
