# Peer review of "Preliminary Evaluation of Pentacene Field Effect Transistors with Polymer Gate Electret as Ionizing Radiation Dosimeters"

_applsci, doi:10.3390/app112311368_

Round 1

Reviewer 1 Report

The article is relevant, original and the results are well supported.

It is suggested to improve the abstract in order to highlight the importance of the paper and make it more concise.

At the beginning of the article I do not understand why the author puts all the sections of the article.

Reviewer 2 Report

The paper presents the development of a device based on field-effect transistors with polymer gate electret for radiation dosimetry measurements. The results showed a clear response to exposure to two different radiation sources.

The response presented linear dose-dependence. The paper shows a very interesting potential of dosimetry usage based on solid-state devices.

Minor changes are requested:

  1. Change dose-units to cGy where rad or krad was written.
  2. Page 7 line 2: Add "electron-positron pair production" after the word: "photoelectron".

Reviewer 3 Report

Preliminary evaluation of pentacene field effect transistors with polymer gate electret as ionizing radiation dosimeters.

Topic is interesting and suitable for this journal. But it is better to check the following points:

  1. The function of Figure 1 needs to be further explained.
  2. Please refer to the optimization in this article compared to previous works in detail.
  3. Explain how the sample was fabricated and measured in more detail.
  4. The explanation for Figure 4 should be more complete.

Round 2

Reviewer 3 Report

The changes have been made correctly, and I agree with publishing the manuscript.